

# Validating osteological correlates for the hepatic piston in the American alligator (*Alligator mississippiensis*)

Clinton A. Grand Pré[1], William Thielicke[2], Raul E. Diaz Jr[3], Brandon P. Hedrick[4], Ruth M. Elsey[5,6] and Emma R. Schachner[7]

[1] Cell Biology and Anatomy, Louisiana State University Health Sciences Center, New Orleans, LA, USA
[2] OPTOLUTION Messtechnik GmbH, Loerrach, Germany
[3] Department of Biological Sciences, California State University Los Angeles, Los Angeles, CA, USA
[4] Department of Biomedical Sciences, Cornell University, Ithaca, NY, United States of America
[5] Louisiana Department of Wildlife and Fisheries, Grand Chenier, LA, USA
[6] Murfreesboro, TN, USA
[7] Department of Physiological Sciences, College of Veterinary Medicine, University of Florida, Gainesville, FL, USA

Corresponding authors
Clinton A. Grand Pré,
cgran9@lsuhsc.edu
Emma R. Schachner,
eschachner@ufl.edu

## ABSTRACT

Unlike the majority of sauropsids, which breathe primarily through costal and abdominal muscle contractions, extant crocodilians have evolved the hepatic piston pump, a unique additional ventilatory mechanism powered by the diaphragmaticus muscle. This muscle originates from the bony pelvis, wrapping around the abdominal viscera, extending cranially to the liver. The liver then attaches to the caudal margin of the lungs, resulting in a sub-fusiform morphology for the entire "pulmo-hepatic-diaphragmatic" structure. When the diaphragmaticus muscle contracts during inspiration, the liver is pulled caudally, lowering pressure in the thoracolumbar cavity, and inflating the lungs. It has been established that the hepatic piston pump requires the liver to be displaced to ventilate the lungs, but it has not been determined if the lungs are freely mobile or if the pleural tissues stretch ventrally. It has been hypothesized that the lungs are able to slide craniocaudally with the liver due to the smooth internal ceiling of the thoracolumbar cavity. We assess this through ultrasound video and demonstrate quantitatively and qualitatively that the pulmonary tissues are sliding craniocaudally across the interior thoracolumbar ceiling in actively ventilating live juvenile, subadult, and adult individuals ($n = 7$) of the American alligator (*Alligator mississippiensis*) during both natural and induced ventilation. The hepatic piston is a novel ventilatory mechanism with a relatively unknown evolutionary history. Questions related to when and under what conditions the hepatic piston first evolved have previously been left unanswered due to a lack fossilized evidence for its presence or absence. By functionally correlating specific characters in the axial skeleton to the hepatic piston, these osteological correlates can be applied to fossil taxa to reconstruct the evolution of the hepatic piston in extinct crocodylomorph archosaurs.

## INTRODUTION

Non-avian sauropsids primarily utilize their costal and abdominal musculature to power their breath cycles (*Brainerd & Owerkowicz, 2006*). Crocodilians developed an additional method for ventilation utilizing the novel diaphragmaticus muscle termed the hepatic piston pump (*Boelaert, 1942*; *Carrier & Farmer, 2000*; *Claessens, 2004*; *Claessens, 2009*; *Uriona & Farmer, 2008*). The diaphragmaticus muscle originates from the pelvis, attaches to the pubic apron and gastralia ventrally, and travels cranially, inserting along the liver capsule and pericardium (Figs. 1A and 1B). When the diaphragmaticus muscle contracts, it pulls the liver and viscera caudally, serving to generate negative pressure in the lungs and the cranial part of the thoracolumbar cavity, inducing inspiration (*Claessens, 2009*; *Farmer & Carrier, 2000b*; *Gans & Clark, 1976*). Exhalation primarily occurs *via* passive relaxation of the diaphragmaticus and/or active contraction of the rectus abdominis muscle and costal musculature resulting in the liver and all viscera sliding cranially (*Claessens, 2009*; *Farmer & Carrier, 2000b*; *Gans & Clark, 1976*; *Uriona & Farmer, 2008*).

The diaphragmaticus muscle is not homologous to the mammalian diaphragm. The mammalian diaphragm is innervated by the phrenic nerves of the cervical plexus and the diaphragmaticus is innervated by spinal nerves 22 and 23. Further, the rectus abdominus and diaphragmaticus muscles are used synchronously for diving, and most likely evolved from the same abdominal muscle group primarily for use in aquatic environments to control pitch and roll during diving in *A. mississippiensis* by shifting the center of mass (*Carrier & Farmer, 2000*; *Uriona & Farmer, 2006*). The evolutionary origin of the hepatic piston has been hypothesized to have been associated with the secondary aquatic habitat of crocodilians, placing the evolution of the diaphragmaticus muscle after the split between Pseudosuchia and Avemetatarsalia, making it distinct from the respiratory anatomy found in non-avian dinosaurs and birds (*Claessens & Vickaryous, 2012*; *Uriona & Farmer, 2008*).

Evaluating the occurrence of soft tissue structures in the fossil record is difficult and best done using osteological correlates; bony anatomical structures or features that can be morphologically linked to soft tissue in extant taxa (*Lauder, 1995*; *Witmer, 1995*). The shift from a terrestrial to a semi-aquatic or aquatic niche in extinct and extant crocodilians is associated with a series of correlates related to cranium shape, body size, or derived from the hindlimb and pelvis as they relate to locomotion (*Benson & Butler, 2011*; *Chamero, Buscalioni & Marugán-Lobón, 2013*; *Hedrick, Schachner & Dodson, 2022*; *Hua, 2003*; *Iijima, Kubo & Kobayashi, 2018*; *Iijima & Kubo, 2019*; *Mannion et al., 2015*; *Molnar et al., 2015*; *Nesbitt, 2011*; *Parrish, 1986*; *Parrish, 1987*; *Salisbury & Frey, 2000*; *Salisbury et al., 2006*; *Stockdale & Benton, 2021*; *Sullivan, 2015*; *Wilberg, 2015*; *Wilberg, Turner & Brochu, 2019*; *Young et al., 2012a*; *Young et al., 2012b*). For example, elongate limbs, dorsoventrally tall crania, and well-developed fourth trochanters of femora are associated with terrestrial habitats while paddle-like limbs and a hypocercal tail fin are associated with fully marine environments (*Wilberg, 2015*; *Wilberg, Turner & Brochu, 2019*; *Young et al., 2012a*; *Young et al., 2012b*).

The smooth interior ceiling of the thoracolumbar cavity in *A. mississippiensis* is associated with the following distinct osteological characteristics (Fig. 2): (1) the parapophysis
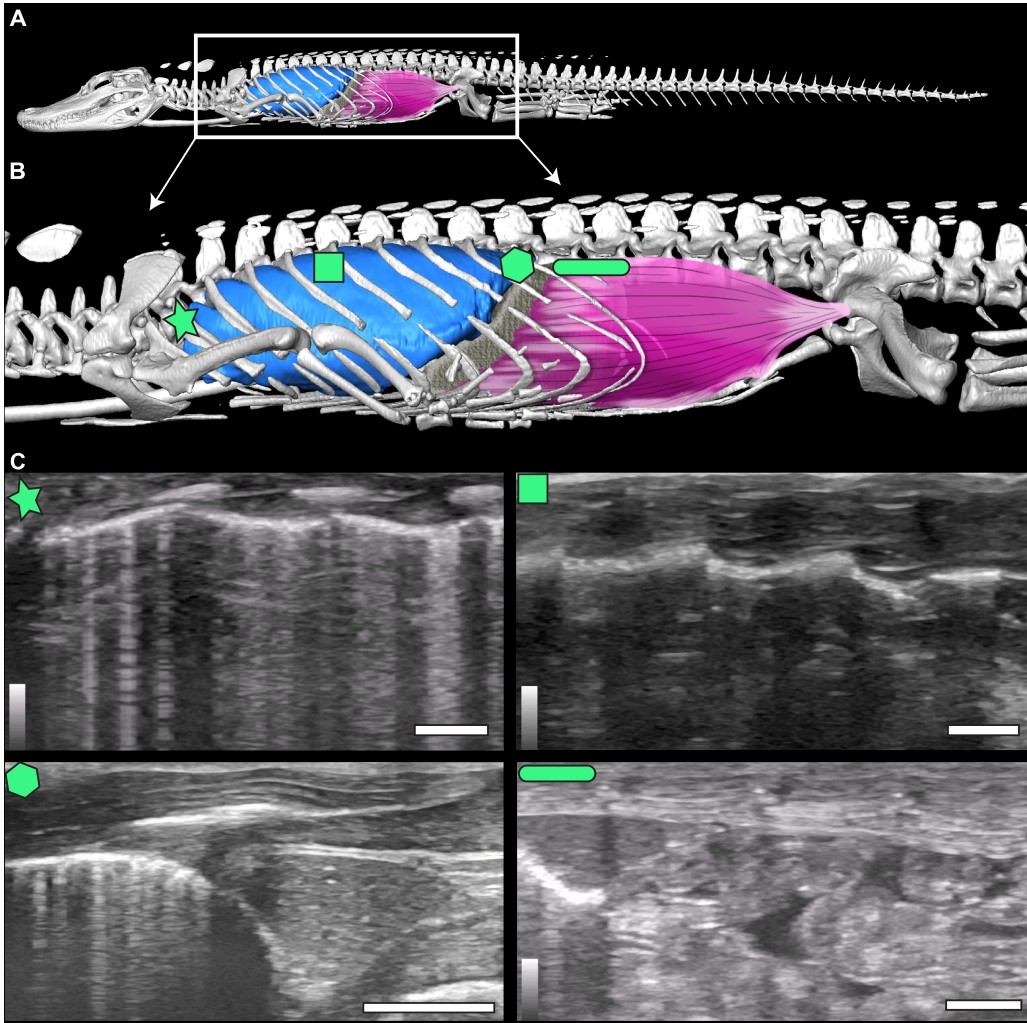

**Figure 1** **μCT model and ultrasound positions.** (A) μCT of *Alligator mississippiensis* (Hamilcar Barca) with a diagrammatic illustration of diaphragmaticus muscle (pink) and liver (gray) in left lateral view. White box represents the region enlarged in (B) which indicates the areas that ultrasound data were collected, represented from cranially to caudally: axillary (star), mid-thoracic (square), hepatic margin (hexagon), and post-hepatic (rectangle). (C) The ultrasound images are oriented in left lateral view with the most cranial region on the left edge of each image. The top of the video is the most superficial and the bottom the deepest. The brightest features in the videos are the ribs in white and the lung pleura running craniocaudal deep to the ribs. The pleura and ribs are visible in the axillary and mid-thoracic images. In the hepatic margin image, the liver is large, triangular in shape, and located in the center-right (caudal) region of the image. It is connected to the bright white lung pleura. In the post-hepatic image, the liver is in the far left (cranial) portion of the image and is connected to the diaphragmaticus muscle which runs craniocaudal. Deep to the diaphragmaticus muscle is the gastrointestinal system of the alligator.

migrates from the vertebral centrum to the transverse process at or after thoracic vertebra three (T3), and progressively out along the transverse process towards the diapophysis sequentially; (2) the presence of broad flat and progressively elongate transverse processes; (3) a lack of forked ribs beyond T3 that create a distinctly smooth dorsal ceiling; and, (4) a rib-free lumbar region. We propose that these features defining a smooth thoracolumbar

ceiling (Fig. 3) allow the lungs, liver, and viscera to slide unimpeded, and are therefore osteological correlates for the hepatic piston pump. Since the diaphragmaticus muscle has been proposed to be primarily a muscle that evolved for an aquatic environment (*Uriona & Farmer, 2008*), these osteological correlates may be useful in identifying when the hepatic piston first evolved and if this evolutionary event coincided with when the ancestors of extant crocodilians moved into semi-aquatic and/or aquatic habitats (*Brocklehurst, Schachner & Sellers, 2018*; *Schachner, Lyson & Dodson, 2009*; *Schachner et al., 2011*). Despite this hypothesis, these correlates have yet to be functionally validated in extant crocodilians. Here, using ultrasound data for seven individuals of *A. mississippiensis* across a growth series (juvenile to adult), we qualitatively and quantitatively demonstrate a functional relationship between a smooth interior thoracolumbar ceiling and the hepatic piston and recommend a distinct set of morphological characters as osteological correlates that can be used to reconstruct the evolutionary history of the hepatic piston in the fossil record.

## MATERIALS AND METHODS

### Anatomical modeling and Microcomputed Tomography ($\mu$CT)

To illustrate the diaphragmaticus muscle, a $\mu$CT scan of one sub-adult 3.4 kg *A. mississippiensis* (identified as Hamilcar Barca for this project) was obtained from a privately owned facility (Scales and Tales Utah) for clinical purposes (unrelated to this study) and donated for this project. The entire alligator was scanned in a prone position, during a natural apnea, at a resolution of two mm slices (80 kVp, 60 mA). The skeleton and lungs of this individual were segmented into a surface model using Avizo 7.1 (Thermo Fisher Scientific) and followed the methods of *Lawson et al. (2021)* and *Schachner et al. (2023)*. The diaphragmaticus muscle and liver were then illustrated on to the model using Adobe Photoshop (Figs. 1A and 1B).

### Ultrasound data collection

A total of 170 individual 16-second ultrasound videos were collected from three juvenile, two sub-adult, and two adult *A. mississippiensis* ($n = 7$) (see Table 1 for the names and snout-vent lengths [SVL] of the specimens). Individuals were scanned at Louisiana State University Health Sciences Center (LSUHSC) in New Orleans and at the Rockefeller Wildlife Refuge in Grand Chenier, Louisiana. Two individuals were part of the collection of Dr. Raul Diaz (then Southeastern Louisiana University) and five individuals were collected by Dr. Ruth Elsey and staff (Rockefeller Wildlife Refuge). All research conducted on living individuals was approved by the LSUHSC Institutional Animal Care and Use Committee (IACUC #6341).

The animals were not sedated and were scanned in an at-rest, prone, stationary position. While at rest, alligators can enter prolonged periods of apnea. Recordings were taken both during natural breathing, when the individual was compliant and breathed deeply on its own, and during induced breathing. Induced breathing was achieved by giving individuals a 5% $CO_2$, balance $N_2$ gas following the procedures outlined in *Douse & Mitchell (1992)* and *Uriona & Farmer (2006)*. Induced breathing reduced the duration of the apneas and allowed for more breaths to be recorded within the video duration. A recording time of

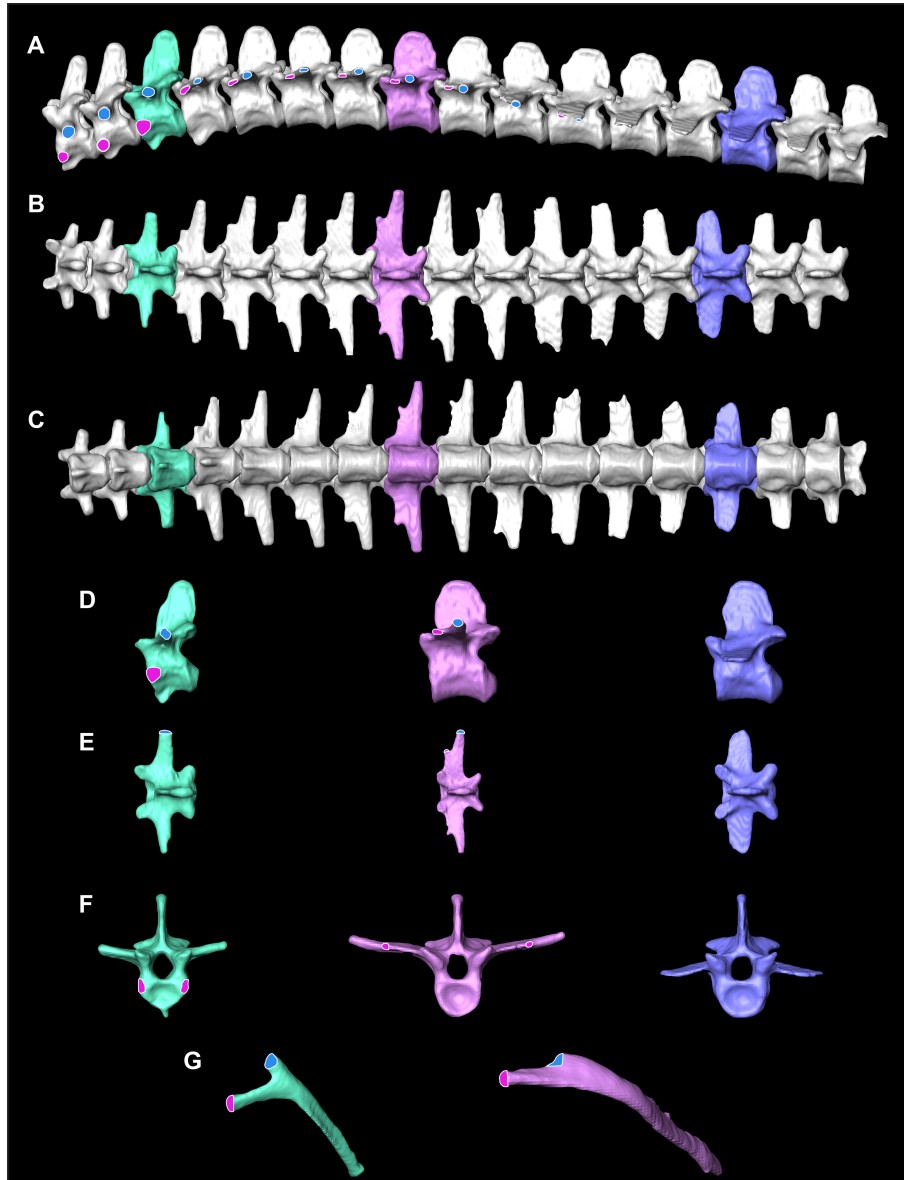

**Figure 2** **Osteological correlates constructed from segmented models of μCT scans of *A. mississippiensis* (Hamilcar Barca).** On T1–T3 the parapophyses (pink circles with white lines) are located on the centra of the vertebrae and the diapophyses (blue circles with white outlines) are located on the transverse processes. Starting on T4 the parapophysis is located on the transverse process cranial to the diapophysis. This pattern continues to T11, the last thoracic vertebra. T3 is highlighted in mint green and exemplifies the forked rib morphology. T8 is in pink and exemplifies the smooth flat transverse process with smooth rib morphology. L3 in blue exemplifies the lumbar vertebrae. (A) Complete thoracic and lumbar vertebral column in left lateral view. (B) Complete thoracic and lumbar vertebral column in dorsal view. (C) Complete thoracic and lumbar vertebral column in ventral view demonstrating the smooth, flat thoracic ceiling—the result of the parapophyses joining the diapophyses on the transverse process. (D) T4, T8 and L3 in left lateral view. (E) T4, T8 and L3 in dorsal view. (F) T4, T8 and L3 in cranial view. (G) Rib 3 in cranial view highlighting the forked shape and Rib 8 in cranial view demonstrating the smooth flat shape.

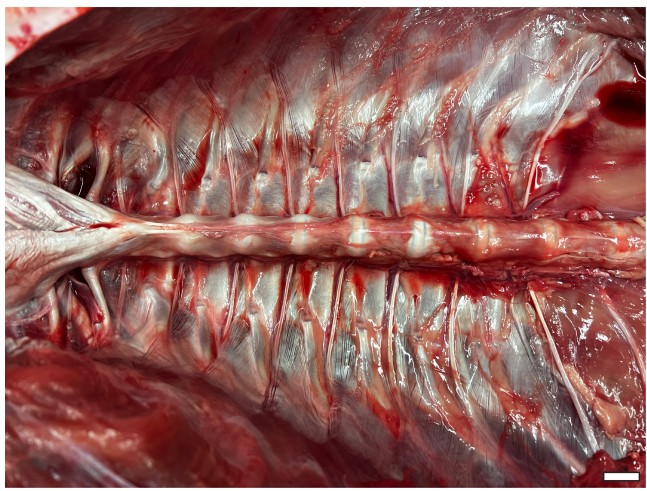

**Figure 3  Thoracic ceiling of *A. mississippiensis*.** All viscera have been removed, demonstrating the smooth interior formed by the morphology of the axial skeleton. Cranial is to the left side of image. Scale bar = two cm.

**Table 1  Specimen data used in this study.**

| Name of Specimen | Life Stage | SVL (cm) | Total Length (cm) |
|---|---|---|---|
| Hannibal Barca | Juvenile | 23 | 57 |
| James Bond | Juvenile | 23 | 58 |
| Scipio Africanus | Juvenile | 31 | 61 |
| Gaius Claudius Nero | Sub-Adult | 56 | 137 |
| Marcus Claudius Marcellus | Sub-Adult | 71 | 142 |
| Archimedes of Syracuse | Adult | 89 | 183 |
| Quintus Fabius Maximus | Adult | 91 | 185 |

16 s was selected to allow adequate time for the animal to take several breaths. Gas was pumped through a polyethylene tube to a bag that was fitted over the nares and mouth of each alligator. Individuals did not breathe the 5% $CO_2$, balance $N_2$ for longer than three minutes at a time. All measurements were taken with the GE LOGIQe Ultrasound using a 12-L probe and penetration depths ranged from two cm for the juveniles to seven cm for the adults. The measurements were collected at four locations (axillary, mid-thoracic, hepatic, and post-hepatic) on the left lateral side of the animal in the coronal plane using consistent anatomical landmarks (Fig. 1C). Individual animals were named for ancient military figures from Carthage, the Roman Republic, and one fictional British MI6 agent for identification.

## Ultrasound measures

Displacement was quantified first by visually establishing the most cranial and most caudal positions of the pleural-hepatic margin during the breath cycle in each video. Frames

  

corresponding with the extremes of the breath cycle were then exported into Adobe Illustrator (Adobe Inc., San Jose, CA, USA) and superimposed. The distance between the most cranial and most caudal positions of the cranial hepatic margin was measured by drawing a line segment between them, the length of which was recorded and scaled to the image size using the scale bars in the ultrasound videos. To ensure consistency, this process was repeated three times and averaged for each video. Results were reported both as total length in centimeters and as a percentage of SVL. This technique, while accurate, could be used only at the hepatic margin position where clear anatomical landmarks could be identified visually.

### PIVlab

To assess movement at other locations where visual assessment and direct measurement were not possible, we utilized a modified version of PIVlab (*Thielicke & Sonntag, 2021*), an open-source program for digital particle image velocimetry (DPIV). DPIV is an analysis technique that can be used to map the spatially resolved velocity or displacement of particle images. DPIV is increasingly used in research areas other than fluid dynamics such as measuring the displacement of diverse textures in digital image data (*Thielicke & Sonntag, 2021*). In the modified version of PIVlab, one or multiple user-generated region of interest (ROI) rectangles were created of any size in any position on the video. Once the ROI was selected, pre-processing image enhancements, Contrast Limited Adaptive Histogram Equalization–CLAHE (*Pizer et al., 1987*) and background subtraction (*Thielicke & Sonntag, 2021*) were performed to increase the signal-to-noise ratio. Cross-correlation was performed in the spatial domain (direct cross-correlation), and as repeated correlation, where the correlation matrices of slightly shifted evaluation windows were multiplied to maximize the signal-to-noise ratio. The displacement of the ROI was determined frame by frame and the displacement was used to offset the ROI in the following frame, enabling the user to lock on the selected pixel pattern in the ultrasound images over time.

Of 170 videos recorded, we were able to measure displacement in 53 videos using PIVlab. The remaining videos could not be processed due to issues with the ultrasound (*e.g.*, dark shadows deep to bone that interfered with the program's ability to track movement) or due to animal movement. It was also evident that tracking boxes frequently lagged behind the tissue in the target window in successful videos, sometimes dramatically. Although these videos demonstrated tissue movement, they underestimated the amount of movement.

## RESULTS

Ultrasound data (see Supplemental Information for ultrasound videos) demonstrate the craniocaudal transition of the pulmonary pleura and liver during breathing relative to the vertebrae at all four measured positions (axillary, mid-thoracic, hepatic, and post-hepatic) in all seven individuals (examples of each location are available in Videos S1–S8). The amount the pulmonary pleura displacement differs at the four positions of measurement. In the axillary position (just caudal and slightly dorsal to the forelimb) the pulmonary pleura can be observed sliding just deep to the vertebrae and ribs in small amounts (Video S1). In the mid-thoracic position (halfway between the pleura/hepatic margin and the axilla), the

pulmonary pleura can be observed sliding deep to the thoracic vertebrae and displacing slightly more than in the axillary position. There are no clear anatomical landmarks that distinguish relative positions of pleura at these locations, so only qualitative observations about the distance of displacement can be made, but the ultrasound videos demonstrate that displacement increases moving caudally (Videos S1–S8). At the hepatic position the pleura is visualized attaching to the cranial aspect of the liver and the diaphragmaticus muscle is attaching to the caudal aspect of the liver. At this position the ultrasound data demonstrate that the displacement is larger than at the mid-thoracic or axillary positions (Videos S4–S7). At the hepatic margin during inhalation, contraction of the diaphragmaticus is visualized, which causes the liver and lung pleura to be pulled caudally. During exhalation, the diaphragmaticus relaxes and the liver and the pleura move cranially in tandem (Videos S4–S7). At the post-hepatic position the diaphragmaticus and the GI can be visualized moving with the liver and the lung pleura (Video S8). At the hepatic margin, the entire contents of the pleuroperitoneal cavity are visualized moving together (Videos S1–S8).

The translation of the viscera is tracked *via* PIVlab at the hepatic position where the lung pleura attaches to the cranial aspect of the liver because the liver is a clear anatomical landmark that interfaces directly with the lung pleura (Fig. 4). Direct measures of displacement are measured at the hepatic margin position for every animal (Table 2). The relationship between the amount of displacement and the use of 5% $CO_2$, balance $N_2$ is not significant suggesting that induced breaths and natural breaths are similar in displacement (Fig. 5A). The amount of displacement varies both between breaths in each individual and between individuals. Average displacements for each individual ranges between 2.90% SVL in Claudius and 8.39% SVL in Hannibal (Table 2). The greatest displacements relative to SVL are measured in the smallest individuals (15.78% of the SVL in Hannibal, 11.11% in Bond). The smallest displacements relative to SVL are in mid-sized to large alligators (Scipio, 1.07%, SVL 31 cm; Claudius, 1.46%, SVL 56 cm; Archimedes, 1.54%, SVL 89 cm). The relationship between the amount of displacement and SVL is not significant and does not scale with size (Fig. 5B).

## DISCUSSION

### Measuring displacement

The main aim of this work was to evaluate if the lung pleura was moving across the smooth interior ceiling of the thoracolumbar cavity during ventilation with the hepatic piston, and if so, to establish associated osteological correlates for this ventilatory mechanism in *A. mississippiensis*. The lung pleura sliding against the smooth ceiling of the thoracolumbar cavity was evident in all videos where deep breaths occurred, with maximum liver displacement ranging from 4.24% to 15.78% SVL. It is likely this study does not capture maximum inspiratory capacity nor the full range of displacement possible under varying physiological conditions. *A. mississippiensis* likely breathes more deeply during or after exercise or while diving (*Farmer & Carrier, 2000a*; *Munns et al., 2005*; *Uriona & Farmer, 2006*; *Uriona & Farmer, 2008*). Additionally, four of seven individuals were wild caught and

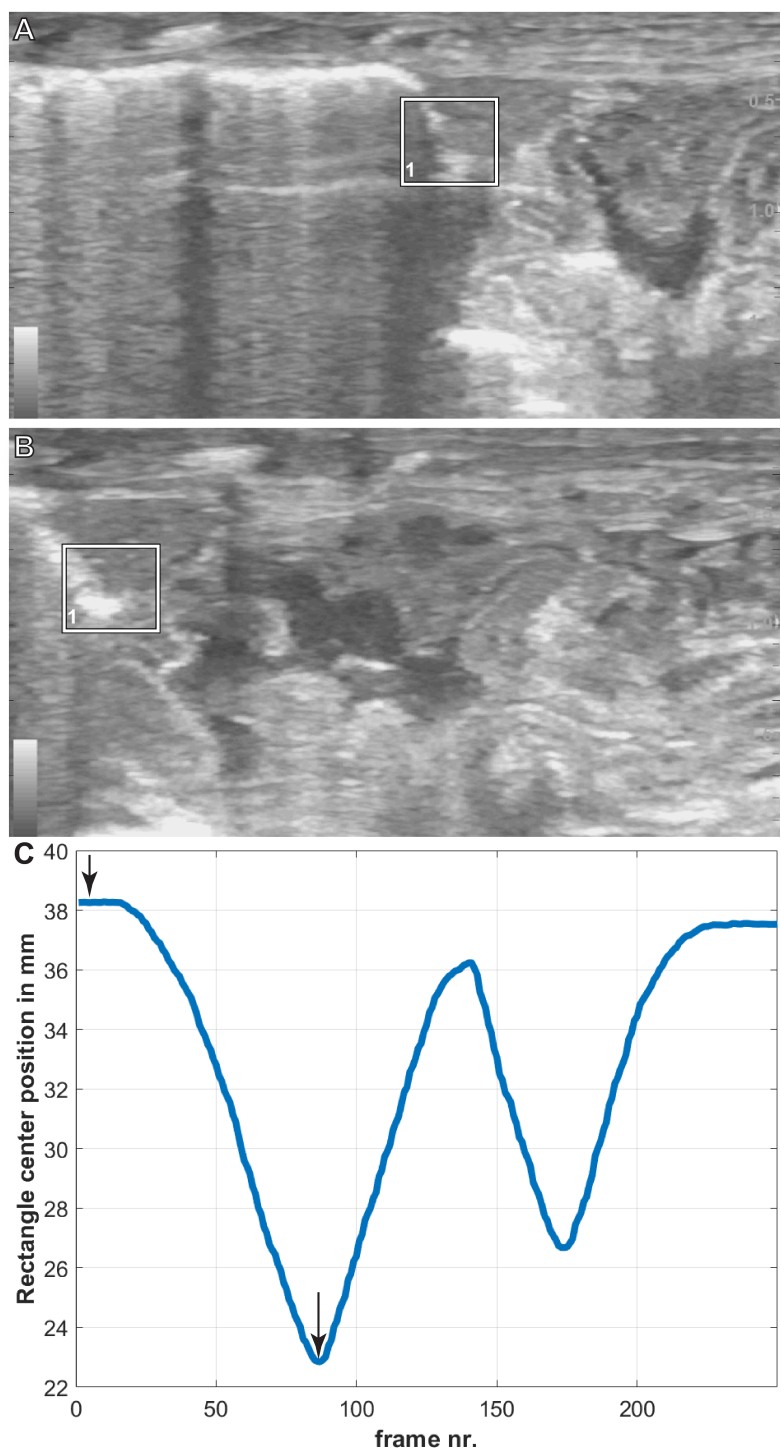

**Figure 4** **Breath cycles in *A. mississippiensis* (James Bond) measured in PIVlab.** (A) PIVlab selection rectangle over the area of interest of the hepatic margin on the cranial edge of the liver in the caudalmost position. (B) PIVlab selection rectangle over the area of interest of the hepatic margin on the cranial edge of the liver in the cranialmost position. (C) The corresponding breath cycle as measured in PIVlab. The black arrows correspond to image A and B from left to right, respectively.

**Table 2  Manually acquired displacement measures at the hepatic margin as percent of SVL.** The *A. mississippiensis* specimens are listed in order of increasing SVL lengths. Values in bold indicate ultrasound video was taken while the animal was breathing the 5% $CO_2$, balance $N_2$ gas.

| Name | SVL (cm) | Displacement Percent (%) | Smallest Recorded Displacement (%) | Greatest Recorded Displacement (%) | Average Displacement (%) |
|---|---|---|---|---|---|
| Hannibal Barca | 23 | **5.58** | 5.58 | **15.78** | 8.39 |
| | | **5.79** | | | |
| | | **5.93** | | | |
| | | **8.88** | | | |
| | | 15.78 | | | |
| James Bond | 23 | 2.19 | 2.19 | 11.11 | 8.01 |
| | | 6.01 | | | |
| | | **7.12** | | | |
| | | 7.80 | | | |
| | | 9.27 | | | |
| | | **9.88** | | | |
| | | **10.73** | | | |
| | | 11.11 | | | |
| Scipio Africanus | 31 | 1.07 | 1.07 | 4.25 | 3.02 |
| | | 2.39 | | | |
| | | 3.25 | | | |
| | | 3.38 | | | |
| | | 3.79 | | | |
| | | 4.25 | | | |
| Gaius Claudius Nero | 56 | 1.46 | 1.46 | **4.25** | 2.90 |
| | | **2.24** | | | |
| | | **3.64** | | | |
| | | **4.25** | | | |
| Marcus Claudius Marcellus | 71 | **4.65** | 4.65 | **9.88** | 6.75 |
| | | 5.87 | | | |
| | | 6.54 | | | |
| | | 6.79 | | | |
| | | **9.88** | | | |
| Archimedes of Syracuse | 89 | 1.54 | 1.54 | **4.24** | 2.94 |
| | | **2.46** | | | |
| | | **3.55** | | | |
| | | **4.24** | | | |
| Quintus Fabius Maximus | 91 | 5.17 | 5.17 | 7.94 | 5.99 |
| | | 5.34 | | | |
| | | **5.52** | | | |
| | | 7.94 | | | |

thus were not fasted before analysis: Eating a large meal (>15% of body mass) can decrease the animal's vital capacity by up to 25% (*Uriona & Farmer, 2006*), which would potentially decrease the maximum displacement measured in these wild individuals. However, it is unlikely these wild-caught individuals ate a large meal at the time of capture, they were

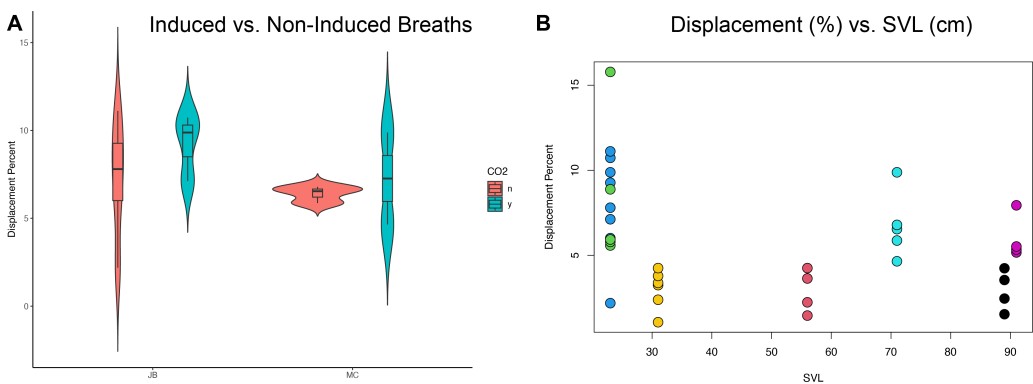

**Figure 5  Statistical analyses of induced breaths and displacement related to size.** (A) Descriptive plot for James Bond (JB) and Marcus Claudius Marcellus (MC) illustrating the lack of a relationship between induced breaths (blue), breathing under normal atmospheric conditions (red), and displacement. (B) The trend is not significant between displacement (%) and SVL (cm). Each color represents a different specimen: Green is Hannibal Barca, blue is James Bond, yellow is Scipio Africanus, red is Gaius Claudius Nero, teal is Marcus Claudius Marcellus, black is Archimedes of Syracuse, and purple is Quintus Fabius Maximus. Specimens can also be differentiated by their SVL and Displacement Percent.

caught during the fall season when feeding frequency decreases substantially (*Lance, 2003*). Other external physiological variables that may have an effect on inspiratory capacity include ambient temperature (*Munns et al., 2012*), if the animal is on land or in water (*Gans & Clark, 1976*; *Uriona & Farmer, 2006*), and if the animal is standing (*Uriona & Farmer, 2006*).

There are also many other internal physiological variables that can lead to substantial changes in total displacement *via* the hepatic piston. Crocodilians can switch between costal/abdominal dominant or hepatic dominant ventilation, greatly varying the amount the hepatic piston contributes to inspiration (*Brainerd & Owerkowicz, 2006*; *Brocklehurst et al., 2017*; *Claessens, 2009*; *Farmer & Carrier, 2000b*; *Gans & Clark, 1976*; *Munns et al., 2012*). Further, crocodilians can enter long periods of apnea (*Gans & Clark, 1976*; *Uriona & Farmer, 2006*), and can increase their total tidal volume (rather than breathing frequency) during periods of hypercapnia (*Claessens, 2009*; *Wang & Warburton, 1995*). Due to the propensity for apnea, we used 5% $CO_2$, balance $N_2$ gas to induce deep breaths (*Douse & Mitchell, 1992*; *Uriona & Farmer, 2006*) but did not find a statistical relationship between the amount of displacement and the use of 5% $CO_2$, balance $N_2$ gas (Fig. 5A). If breathing the 5% $CO_2$, balance $N_2$ gas was putting the individuals in a state of hypoxia we would expect to see an increase in displacement when the individuals breathed the 5% $CO_2$, balance $N_2$ gas. Instead, we observed no measurable difference when animals breathed the gas or did not. Nevertheless, the ability for crocodilians to change the dominant muscles of inspiration and change the frequency and intensity of apnea suggests that the displacement measured does not represent the displacement possible under various internal conditions.

Crocodilians can also change the tension of the abdominal wall and gastralia (*Farmer & Carrier, 2000b*; *Gans & Clark, 1976*), have a rotating pubis that changes the potential caudal extent of visceral displacement (*Claessens, 2004*; *Claessens, 2009*; *Farmer & Carrier, 2000b*),

and can maintain or change the pressure gradient on either side of the post-pulmonary septum (PPS) (*Cramberg, Greer & Young, 2022*). *Klein & Owerkowicz (2006)* discuss that the "diaphragmaticus-endowed PPS" can assist with cranial visceral displacement to avoid lung collapse. It should be noted that *Klein & Owerkowicz (2006)* often reference the crocodilian diaphragmaticus and PPS functionally together, but they are distinct anatomical structures (*Cramberg, Greer & Young, 2022*). Despite the known ability of crocodilians to drastically change how much the hepatic piston contributes to inspiration, our measures of displacement are consistent with observations from *Schachner et al. (2022)* and *Claessens (2009)*.

μCT is a useful method for evaluating the potential of lung movement and of inspiratory capacity in crocodilians. *Schachner et al. (2022)* imaged multiple specimens of live and deceased Cuvier's dwarf caiman (*Paleosuchus palpebrosus*) in different states: post-prandial, fasting, open to atmosphere, and at total lung capacity. Post-prandial individuals had cranially displaced lungs up to five vertebral segments compared to a fasted state. Individuals of deceased hatchling *P. palpebrosus* that had fully inflated lungs had a caudal displacement of 1.5 vertebral lengths relative to individuals scanned that were open to atmosphere. This demonstrates the large variability in crocodilian lung volume and position and is consistent with the results found here for *A. mississippiensis*. In the ultrasound videos at the pleural hepatic margin the pleura and liver are routinely sliding across one to three vertebral lengths (see supplementary materials on Data Dryad for all ultrasound videos). This is also consistent with *Claessens (2009)*, who reports, using cineradiographic techniques, an average displacement of the cranial surface of the liver of 1.4 vertebral lengths in juvenile individuals of *A. mississippiensis*.

We did not find a correlation between size and displacement (Fig. 5B). *Gans & Clark (1976)* report, using EMG data in *Caiman crocodilus*, that breath frequency decreases in larger animals. This study includes 20 individuals that range in size from 400 g to 7.5 kg. *Claessens (2009)* in a cineradiographic study of five juvenile *A. mississippiensis*, ranging from 55–115 cm total length, reported liver displacement is larger in larger individuals. *Munns et al. (2012)* reported personal observations that the diaphragmaticus muscle in juvenile crocodilians are thin and translucent and thicker and more well-developed in adults. We confirm this observation (pers. obs., CGP and ERS) in dissections of *A. mississippiensis*. The limitations of correlating size and displacement include that *Claessens (2009)* and *Gans & Clark (1976)* studies contain no adults. Further, all studies, including this one, are limited in that crocodilians are known to vary their reliance on the hepatic piston during ventilation, therefore capturing the total variation is difficult for any one study. This is not a physiologic study on all possible variation in the displacement *via* the hepatic piston. To vigorously study the relationship between size and hepatic piston use we suggest a mixed methodology that includes XROMM and electromyography on many individuals ranging in size from hatchling to adults in numerous physiologic conditions.

## PIVLAB

PIVlab was useful in providing clear respiration curves (Fig. 4). Our measurements of displacement in PIVlab underestimated displacement compared to the manual

measurements at the hepatic margin. This is not the result of an inherent flaw with PIVlab but is related to the nature of ultrasound data. Ultrasound data from the thoracic region commonly have shadows from ribs that interrupt tracking in PIVlab and have lower contrast between the target and background ultrasound scatter. The human eye and manual tracking can follow movement that an automated program cannot. Also, PIVlab functions by maintaining the tracking box in every frame of the video. A shadow that may darken halfway through the video is enough for PIVlab to lose its target: The PIVlab algorithm tracks the most statistically significant pixel pattern. If a shadow dominates the tracked region, then the algorithm will tend to track the shadow. The manual procedure needs a clear frame of only the most cranial and most caudal aspects of the breath to measure the displacement. However, this initial work and prior studies that have used DPIV with ultrasound (*Farron, Varghese & Thelen, 2009*; *Korstanje et al., 2009*; *Korstanje et al., 2010*; *Korstanje et al., 2012*; *Loram, Maganaris & Lakie, 2006*) demonstrate that there is promise in advancing PIVlab technology to work more consistently with ultrasound data.

## Heterogeneity of lung movements

When describing the lung of the Nile crocodile (*Crocodylus niloticus*), *Perry (1988)* states that it will "tend to fuse with the parietal pleural surface" and therefore the lung tissue stretches instead of sliding during inspiration *via* the hepatic piston (particularly the cranial half). There are two main points in *Perry*'s (*1988*) work that are of interest: (1) the *C. niloticus* lung is more strongly attached to the parietal pleura cranially than caudally, and as a consequence, (2) in the caudal half of the lung, the pleural tissue is stretching instead of sliding. It is possible that the specimens evaluated by Perry with fusion between the visceral and parietal pleura were pathologic, but this needs to be investigated further in other adult specimens of *C. niloticus.* We qualitatively demonstrate in this study that the amount of pleural displacement increases moving from cranially to caudally in all individuals *via* the ultrasound videos in *A. mississippiensis*. There is minimal displacement in the axillary position, where the ribs are forked, but the visceral pleural tissues do not appear to be fused with the parietal pleural. The largest amount of pleural displacement is in the hepatic position where the thoracic vertebrae are wide and the ribs are flat, creating a smooth surface (on the interior thoracic ceiling) on which the lung tissue can either expand or slide freely. However, studies on the potential heterogeneous mechanical properties of the lung pleura may reveal more information related to the structural changes in the lung tissue that *Schachner et al. (2022)* and *Perry (1988)* observed and how this varies across Crocodylia. *Schachner, Hutchinson & Farmer (2013)* did not find any fusion between the visceral and parietal pleural described for *C. niloticus* by *Perry (1988)* and there may be intraspecific variation or pathologies associated with fusion observed in some of these animals. It is possible that adhesions occur variably and impact hepatic piston function. Additional investigation into differential movements of the alligator pulmonary tissues will also reveal if there are any functional relationships between differential regional mobility/immobility of the pulmonary pleura and the architecture of the bronchial tree.

### Evolution of the hepatic piston

Of all the various morphologically and ecologically diverse pseudosuchian groups that thrived during the Late Triassic period, only small, terrestrial crocodylomorph pseudosuchians survived the Triassic/Jurassic (Tr-J) mass extinction ~200 million years ago (*Butler et al., 2011*; *Nesbitt, 2011*; *Wilberg, Turner & Brochu, 2019*). These small gracile-limbed reptiles were the terrestrial ancestor to all crown-group semi-aquatic extant crocodilians (*Nesbitt, 2011*). Following the Tr-J extinction event, Crocodylomorpha quickly diversified and these animals were found in a wide array of niches by the Middle Cretaceous, ranging from fully marine to terrestrial localities, and potentially even arboreal habitats (*Mannion et al., 2015*; *Wilberg, Turner & Brochu, 2019*). It has been hypothesized that crocodylomorph archosaurs transitioned from terrestrial to semi-aquatic or to marine environments multiple times (*Lessner et al., 2023*; *Schwab et al., 2020*; *Wilberg, Turner & Brochu, 2019*). Given the hypothesis that the hepatic piston evolved in association with diving by controlling pitch and roll (*Uriona & Farmer, 2008*), establishing when the hepatic piston first evolved may refine our understanding of the complex terrestrial and aquatic evolution of crocodylomorphs. Our proposed osteological correlates for the hepatic piston offer a novel set of postcranial characters that can be used to evaluate this environmental history.

## CONCLUSIONS

Here we establish a functional relationship between previously described osteological correlates in the axial skeleton of the American alligator and the ventilatory movements of the hepatic piston mechanism. The loss of the forked ribs cranially, combined with the broadening and flattening of the transverse processes and a rib-free lumbar region, results in a smooth interior ceiling of the thoracolumbar cavity, allowing the pleura to slide cranially/caudally during ventilation. Visualization of the sliding pleura *via* ultrasound identifies the functional relationship between axial skeletal morphology and pleural displacement, validating these osteological correlates which can be utilized in reconstructing the origin and evolution of the hepatic piston ventilatory mechanism in extinct crocodylomorph archosaurs.

## ACKNOWLEDGEMENTS

We thank Dwayne LeJeune, Mickey Miller, and Nick Latiolais for assistance with capture and handling of alligators used in the study. CG Farmer and Dan Malleske were integral to the primary exploratory phase of this work. Annette Gray and Adam Lawson provided critical assistance with collecting the ultrasound data. Special thank you to Scales and Tails Utah for permission to use a scan of one their animals, and Scott Echols DVM for providing imaging data.

### Funding

Emma R. Schachner and Brandon P. Hedrick received a Louisiana State University Health Sciences Center Research Enhancement Program Grant. The funders had no role in study design, data collection and analysis, decision to publish, or preparation of the manuscript.

### Grant Disclosures

The following grant information was disclosed by the authors:

Louisiana State University Health Sciences Center Research Enhancement Program Grant.

### Competing Interests

Brandon P Hedrick is an Academic Editor for PeerJ. William Thielicke is employed by OPTOLUTION Messtechnik GmbH.

### Author Contributions

- Clinton A. Grand Pré conceived and designed the experiments, performed the experiments, analyzed the data, prepared figures and/or tables, authored or reviewed drafts of the article, and approved the final draft.
- William Thielicke conceived and designed the experiments, analyzed the data, authored or reviewed drafts of the article, and approved the final draft.
- Raul E. Diaz Jr analyzed the data, authored or reviewed drafts of the article, and approved the final draft.
- Brandon P. Hedrick performed the experiments, analyzed the data, prepared figures and/or tables, authored or reviewed drafts of the article, and approved the final draft.
- Ruth M. Elsey performed the experiments, analyzed the data, authored or reviewed drafts of the article, and approved the final draft.
- Emma R. Schachner conceived and designed the experiments, performed the experiments, analyzed the data, prepared figures and/or tables, authored or reviewed drafts of the article, and approved the final draft.

### Animal Ethics

The following information was supplied relating to ethical approvals (*i.e.*, approving body and any reference numbers):

Louisiana State University Health Sciences Center Institutional Animal Care and Use Committee provided full approval of this research.

### Data Availability

The raw data and 8 ultrasound videos are available in the Supplementary Files.

The ultrasound videos are available at Data Dryad: Grand Pre, Clinton A. (2023). American alligator ultrasound and microCT [Dataset]. Dryad. https://doi.org/10.5061/dryad.866t1g1w0

The CT Image Series of the Full Body μct Of An American Alligator is available at Morphosource: 10.17602/M2/M529382.

## Supplemental Information

Supplemental information for this article can be found online at http://dx.doi.org/10.7717/peerj.16542#supplemental-information.

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
