# Peer review of "Validating osteological correlates for the hepatic piston in the American alligator (Alligator mississippiensis)"

_PeerJ, doi:10.7717/peerj.16542_

## Round 0.1 · original submission · Major Revisions

Thank you for submitting your work to PeerJ. I apologize for the delay, but now I have received reviews from 3 reviewers. All reviewers agreed that your work has generally sound results and the overall methodological approach appeared robust. Nevertheless, there were a number of issues pointed out by the reviewers that need to be addressed in your revision. In particular, reviewer 1 indicated a series of concerns regarding the displacement measurement illustrated in Figure 4 that need attention. Reviewer 2 has suggested a number of important references that should be given their credit and explained as it relates to your work. Finally, reviewer 3 has highlighted a crucial point on the methods as to what exactly “at rest” meant to the authors, which potentially represents a major source of variation that needs to be addressed. These major points, and many related issues, particularly from reviewer 3, should be dealt with in full during the revision. Thank you for your time.

Reviewer 1 ·

Basic reporting

This paper deals with the ventilation mechanics of the American alligator, and particularly on the interactions between the lung and the 'smooth thoracic ceiling' formed by the modified axial skeleton of crocodilians.

The article is well structured, and the tables look good. The anatomical models and photographs in the figures are very high quality. I would prefer to see the structures in the ultrasound data (e.g., Figures 4, 5) outlined rather than painted over. The authors have shared a good amount of raw data, but the accompanying metadata needs to be improved - for example, it is not clear which animals feature in the supplementary videos provided. Also, will all the ultrasound videos measured in the provided supplemental table be made available post-publication? Or only those currently provided as part of the supplement?

Some aspects of the study could be better introduced. For example, the non-significant relationship between percentage displacement and snout-vent length is interesting as several studies have previously proposed that crocodilians increasingly rely on the hepatic piston (and so we might see greater % displacement) at larger body sizes (e.g., Gans and Clark, Uriona and Farmer, Munns et al). However, this particular point is never raised by the authors, and I think represents a missed opportunity. Similarly, regarding induced breaths, was this just to get the animals to breathe at all, or were you trying to deliberately induce deep breathing? Why is it important that induced or non-induced breaths are (not) significantly different?

There are additional literature references the authors have not cited, and which I would like to see discussed. For example, in his detailed work on the anatomy of the crocodilian lung, Perry 'suggests that the lungs do not slide, but rather stretch to accompany liver movement'. This statement is directly contradicted by the authors' experimental data and deserves mentioning, at least in the introduction to set up the previous uncertainty on how the lungs interact with the skeleton.

I realize that the pulmonary-hepatic boundary was the easiest anatomical marker to track, and so the caudal region of the lungs was the easiest to measure displacement for. However, it has previously been shown that the crocodilian lung has a heterogeneous structure, and so different regions of the lung might expand or contract to differing degrees during ventilation. Since the authors recorded different parts of the lung (illustrated in Figure1), can they comment on regional differences in lung motion or displacement at all?

I have additional one note on the language of the paper: would it be possible to shorten the given names of the animals in the main text, and only use the full names in the supplementary table? I think this would improve both flow of the text and clarity for the reader.

Additional References:
Munns SL, Owerkowicz T, Andrewartha SJ, Frappell PB. 2012. The accessory role of the diaphragmaticus muscle in lung ventilation in the estuarine crocodile Crocodylus porosus. The Journal of Experimental Biology 215:845–852. DOI: 10.1242/jeb.061952.

Perry SF. 1988. Functional morphology of the lungs of the Nile crocodile, Crocodylus niloticus: non-respiratory parameters. Journal of Experimental Biology 134:99–117.

Experimental design

The primary research falls within the Aims and Scope of the journal and the research question - how do the lungs move within the thoracic cavity of crocodilians, and how does this relate to the axial skeleton - is well stated and addresses an important gap in our understanding of breathing mechanics within reptiles, with implications for respiratory evolution in fossil taxa. This is the first time, that I am aware of, that ultrasound imaging has been applied to study breathing mechanics in reptiles, outside of a clinical context.

I commend the authors for their use of automated tracking software to facilitate their analyses of cranio-caudal liver displacement - this appears to be a novel application of these techniques in the field of respiratory biomechanics (given that all the references the authors provide lines 253-255 focus on muscle and tendon measurements). However, I have some comments on how the manual measurements were taken. While there is nothing wrong with the approach used here, in future I would advise using a dedicated point tracking or video analysis software for manual measurement, such as Tracker to produce traces similar to those from the automated methos presented in Figure 5.

In my attempts to perform due diligence as a reviewer, I am struggling to replicate the displacement measurement illustrated in Figure 4. It seems as if the authors measured from the caudal-most point when the liver is cranially positioned, but the cranial-most point when the liver is caudally positioned. Is this a mistake in the figure? Why are the measurements not taken from the same point? Also, speaking as someone unfamiliar with ultrasound methods and their outputs, how was the scale factor between image pixels and real-world distance derived?

Validity of the findings

While I have requested clarification on some of the measurements taken by the authors, I have every confidence in the central finding of this study - that the lungs of crocodilians do slide along the thoracic ceiling. This is clearly visible in the ultrasound videos, answers the research question that the authors' put forward at the start of their study, and forms the main basis of the their conclusions.

Reviewer 2 ·

Basic reporting

This is an amazing paper that I read with great interest. Based on a series of previous work from the last author’s group, the authors hypothesized that a smooth thoracic ceiling correlates with the hepatic piston pump, which facilitates the movement of the lung back and forth, therefore changing the CoM, during diving. To test this hypothesis, they conducted ultrasound recording in live American alligators, demonstrating the movement of the lung during breathing. I believe that the osteological correlates of the lung movement established here can provide valuable insights into the origin and evolution of the hepatic piston, a unique ventilation mechanism observed in extant crocodylians. I have only a few minor suggestions.

Experimental design

The experimental design was thoroughly explained, and all raw data will be made available to readers upon acceptance of the paper.

Validity of the findings

The results were extensively shown, and all underlying data will be available upon acceptance of the paper. Conclusions drawn from the results were well stated and aligned with their primary findings.

Additional comments

Line 75: In terms of axial and appendicular adaptations to increased aquatic ecology in crocodyliforms, several additional studies can be cited, including Salisbury and Frey (2000), Hua (2003), Salisbury et al. (2006), Chamero et al. (2013), Iijima et al. (2018), and Iijima and Kubo (2019).

Line 77: Did these ichnology papers discuss gait changes related to the shift from a terrestrial to semiaquatic ecology?

Line 98: A 3.4 kg individual (~3 feet TL?) may not be an adult. Both males and females of American alligator typically reach sexually maturity at 6 feet TL (Vliet 2020).

Line 100: Could you be more specific about the source of the specimen (e.g., name and location of the facility)?

Line 131: Sounds fun!

Line 221: Schwab et al. (2020) on inner ear and Lessner et al. (2023) on trigeminal nerve evolution across crocodylomorphs can be additionally cited.

Line 232: Among extant crocodylians, parapophyseal and diapophyseal facets of a transverse process remain separated posteriorly through the 11th dorsal vertebra or anterior in non-Gavialis taxa, while in Gavialis, they remain separated through the 12th dorsal vertebra (Iijima and Kubo 2019). This position generally corresponds to the presence of the posteriormost rib. Therefore, the rib-free lumber region appears to be smallest in Gavialis, the most aquatic of the extant crocodylian.

References cited in review
Chamero B, Buscalioni ÁD, Marugán-Lobón J. 2013. Pectoral girdle and forelimb variation in extant Crocodylia: the coracoid–humerus pair as an evolutionary module. Biological Journal of the Linnean Society 108:600–618.
Hua S. 2003. Locomotion in marine mesosuchians (Crocodylia): the contribution of the “locomotion profiles.” Neues Jahrbuch für Geologie und Paläontologie - Abhandlungen 227:139–52.
Iijima M, Kubo T. 2019. Comparative morphology of presacral vertebrae in extant crocodylians: taxonomic, functional and ecological implications. Zoological Journal of the Linnean Society 186:1006–25.
Iijima M, Kubo T, Kobayashi Y. 2018. Comparative limb proportions reveal differential locomotor morphofunctions of alligatoroids and crocodyloids. Royal Society Open Science 5:171774.
Lessner EJ, Dollman KN, Clark JM, Xu X, Holliday CM. 2023. Ecomorphological patterns in trigeminal canal branching among sauropsids reveal sensory shift in suchians. Journal of Anatomy 242:927–52.
Salisbury SW, Frey E. 2000. A biomechanical transformation model for the evolution of semi-spheroidal articulations between adjoining vertebral bodies in crocodilians. In: Crocodilian Biology and Evolution Chipping Norton: Surrey Beatty & Sons. p. 85–134.
Salisbury SW, Molnar RE, Frey E, Willis PMA. 2006. The origin of modern crocodyliforms: new evidence from the Cretaceous of Australia. Proceedings of the Royal Society B: Biological Sciences 273:2439–48.
Schwab JA, Young MT, Neenan JM, Walsh SA, Witmer LM, Herrera Y, Allain R, Brochu CA, Choiniere JN, Clark JM, Dollman KN, Etches S, Fritsch G, Gignac PM, Ruebenstahl A, Sachs S, Turner AH, Vignaud P, Wilberg EW, Xu X, Zanno LE, Brusatte SL. 2020. Inner ear sensory system changes as extinct crocodylomorphs transitioned from land to water. Proceedings of the National Academy of Sciences 117:10422–28.
Vliet KA. 2020. Alligators: the illustrated guide to their biology, behavior, and conservation Baltimore: Johns Hopkins University Press.

Reviewer 3 ·

Basic reporting

There needs to be more clarity about exactly what you are hypothesizing and testing. The confusion arises because you are too imprecise when you reference “viscera” (which can occur anywhere in the pleuroperitoneal cavity), “hepatic” which, though mobile, is restricted to the mid and caudal portions of the cavity, and “lung” which, though mobile, is restricted to the rostral and mid portions of the cavity. Your Figure 3 is a good example; illustrating the “thoracic ceiling” is fine, but how does this relate to hepatic displacement? The illustration is the rostral portion of the pleuroperitoneal cavity, which is filled by the lungs but has little to do with the liver (your own reconstruction in Figure 1 places the liver beyond or at the very margin of Figure 3. The confusion for the reader stems in part from the fact that displacement of the lung during ventilation (“sliding” is probably not the ideal term) is a very common feature of terrestrial vertebrates including reptiles. When you look at the XROMM images Brainard shared of Iguana ventilating, you can see the lungs displacing. Iguana, like other terrestrial vertebrates has relatively smooth inner thoracic walls to facilitate lung motion. You don’t provide any comparative context to help your reader assess if the magnitude of lung displacement you found is larger or smaller than in other reptiles. The osteological correlation seem far more pertinent to the displacement of the liver, you cloud that interesting line of inquiry by not separating lung and liver. Lung displacement is everywhere; hepatic displacement is rather unique…focus more on that and the paper will flow much better.

The anatomy (beyond the osteology) is poorly presented, and would mislead the reader. The diaphragmaticus muscle does not “encapsulate” the viscera, nor does it attach to the pericardium. As Cramberg demonstrated clearly in the recent special issue of Anatomical Record, the alligator has a diaphragm separating the liver from the lungs. You may prefer to refer to the diaphragm as the post-pulmonary septum, but since it separates the structures you are studying, and (by also attaching to the body wall) limits the mobility of the system, failing to even note its presence is inexcusable. You note that caudal displacement of the liver would create negative pressures in the “cranial portion of the pleuroperitoneal cavity.” Obviously regional differences in pressure are only possible with the presence of a diaphragm like structure; Cramberg recorded the differential pressures you are postulating.

Experimental design

There is a significant methodological problem that needs to be addressed. You state that all of the breaths were recorded with the alligators “at rest”, but in the context of hepatic displacement what does “at rest” mean? As you clearly represent in the ultrasound images, the liver of the alligator has a characteristic shape. It also has a rather solid texture (like most vertebrate livers). The caudal portion of the pleuroperitoneal cavity of the alligator does not have a static characteristic shape, it is quite dynamic depending on (among other things) the posture of the animal. Simplistically, when an alligator is moving in a “high walk” the pleuroperitoneal cavity narrows in the medial-lateral direction and expands in the dorsal-ventral direction; when the alligator sprawls and rests all of its body weight on the substrate the pleuroperitoneal cavity expands in the medial-lateral direction and narrows in the dorsal-ventral direction. The displacement of the fixed-shape liver is surely influenced by the dynamics of shape change in the pleuroperitoneal cavity. Telling the reader that the alligators were “at rest” during data collection is inadequate to address this, and may well be a significant (and undiscussed) source of variation.

Validity of the findings

The findings are clearly presented. I am concerned about how much influence the methodological problem described above influenced the results. If you can not document the shape of the body, or the relative tension of the body wall, you could be comparing visceral displacement under very different conditions.

Additional comments

1) What possible justification is there for publishing the names you chose to give the animal subjects of your study? Is everyone going to find this as endearing as you do? Gaius Claudius Nero butchered a large number of Carthaginian males, and subjected the females to a lifetime of sexual slavery….which part of that are you celebrating? Follow the norms of scientific publication like everyone else.

2) Your explanation of the PIVlab is excessive. You can delete lines 146-159 without any impact on the reader’s ability to understand what you did.

3) Line 227: “Given that we have shown the lung slides in the pleuroperitoneal cavity along the ventral surface of the vertebrae” …there is a logical error in jumping from this to demonstrating aquatic habitat preference. My lungs are sliding along the ventral surface of my vertebrae as I am writing this.

4) Figure 7 is nice, and I enjoyed it when you first published it. Here it seems too much like an add-on that is unnecessary.

5) A discussion of the relative applicability of PIVlab (lines 243-256) seems like an odd way to conclude a paper seeking skeletal clues for the aquatic origins of crocodylians.

6) The connection between buoyancy control and the aquatic origin of the hepatic pump is not supported. The fact that Alligator can use hepatic displacement to regulate buoyancy does not indicate that the hepatic pump evolved in the aquatic environment.

7) I believe that your line investigation would be strengthened by a broader comparative approach. More specifically, I would suggest a detailed comparison of fossil crocodylians and mosasaurs. Similar size ranges, a similar pattern of repeated transitions between terrestrial and aquatic habitats.....

---

## Round 0.2 · accepted · Accept

All reviewers have agreed that the manuscript has been greatly improved in this revised submission, so I am more than happy to accept your manuscript! Thank you for submitting to PeerJ.

Reviewer 1 ·

Basic reporting

No comment

Experimental design

Thank you for addressing the concerns I raised earlier. I am much happier with the presentation of the methods now.

Validity of the findings

The manuscript is much improved - the methods are better explained, and I think the findings are very robust. The main findings of the paper are better integrated into the current literature. I believe the authors have addressed all of my concerns, and I think the study is now ready for publication.

Reviewer 2 ·

Basic reporting

The authors addressed all the inquiries that I made. I have no further comments.

Experimental design

no comment

Validity of the findings

no comment

Additional comments

no comment

Reviewer 3 ·

Basic reporting

Much improved. The manuscript flows well. Worth publication.

Experimental design

The revisions have improved this aspect of the manuscript.

Validity of the findings

Again, the revision significantly helped clarify the connection between your results and the broader topics you wish to address in the Discussion.

Additional comments

Two things caught my eye when working through the revision:
1) Lines 310-313 ...is the second point stated correctly? It seems like "stretching" and "sliding" are switched.

2) You document differential displacement in different portions of the lung, but you don't really ever address why you think that occurs. You touch on it around 321, but I think most readers would be interested and would like to see more. Is the greater displacement caudally purely an additive phenomenon? Does the differential displacement correlate to the size (or other aspect?) of the defined portions of the lung in Alligator? Is there a relationship between degree of displacement and the airflow pattern through the lung of Alligator?